# Long-Term Impact of Extreme Weather Events on Grassland Growing Season Length on the Mongolian Plateau

Wanyi Zhang [1,2,3], Qun Guo [1,3], Genan Wu [4], Kiril Manevski [2,5,*] and Shenggong Li [1,2,3]

1   National Ecosystem Science Data Center, Key Laboratory of Ecosystem Network Observation and Modeling, Institute of Geographic Sciences and Natural Resources Research, Chinese Academy of Sciences, 11 Datun Road, Chaoyang District, Beijing 100101, China; zhangwanyi22@mails.ucas.ac.cn (W.Z.); guoq@igsnrr.ac.cn (Q.G.); lisg@igsnrr.ac.cn (S.L.)

2   Sino-Danish Center for Education and Research, Eastern Yanqihu Campus, University of Chinese Academy of Sciences, 380 Huaibeizhuang, Beijing 101400, China

3   University of Chinese Academy of Sciences, Beijing 100190, China

4   Institute of Spacecraft Application System Engineering, China Academy of Space Technology, Beijing 100094, China; wugn.15s@igsnrr.ac.cn

5   Department of Agroecology, Aarhus University, Blichers Alle 20, 830 Tjele, Denmark

*   Correspondence: kiril.manevski@agro.au.dk

**Abstract:** Quantifying extreme weather events (EWEs) and understanding their impacts on vegetation phenology is crucial for assessing ecosystem stability under climate change. This study systematically investigated the ecosystem growing season length (GL) response to four types of EWEs—extreme heat, extreme cold, extreme wetness (surplus precipitation), and extreme drought (lack of precipitation). The EWE extremity thresholds were found statistically using detrended long time series (2000–2022) ERA5 meteorological data through z-score transformation. The analysis was based on a grassland ecosystem in the Mongolian Plateau (MP) from 2000 to 2022. Using solar-induced chlorophyll fluorescence data and event coincidence analysis, we evaluated the probability of GL anomalies coinciding with EWEs and assessed the vegetation sensitivity to climate variability. The analysis showed that 83.7% of negative and 87.4% of positive GL anomalies were associated with one or more EWEs, with extreme wetness (27.0%) and extreme heat (25.4%) contributing the most. These findings highlight the dominant role of EWEs in shaping phenological shifts. Negative GL anomalies were more strongly linked to EWEs, particularly in arid and cold regions where extreme drought and cold shortened the growing season. Conversely, extreme heat and wetness had a greater influence in warmer and wetter areas, driving both the lengthening and shortening of GL. Furthermore, background hydrothermal conditions modulated the vegetation sensitivity, with warmer regions being more susceptible to heat stress and drier regions more vulnerable to drought. These findings emphasize the importance of regional weather variability and climate characteristics in shaping vegetation phenology and provide new insights into how weather extremes impact ecosystem stability in semi-arid and arid regions. Future research should explore extreme weather events and the role of human activities to enhance predictions of vegetation–climate interactions in grassland ecosystems of the MP.

**Keywords:** cold; extreme drought; event coincidence analysis; extreme heat; phenology; solar-induced chlorophyll fluorescence; extreme wetness

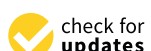

## 1. Introduction

Amid ongoing global change and increased radiative forcing, the probability of weather anomalies and extreme weather events (EWEs) shows a persistent upward trend

everywhere in the world [1–3]. These changes not only alter global climate patterns but also have profound impacts on plant growth cycles, ecosystem productivity, and key ecological processes such as the carbon cycle [4–6]. However, defining extreme weather events such as drought, heat or cold waves, and especially their co-occurrence remains a research challenge due to the inherent climate variability, which is dependent on regional hydrometeorological and pedoclimatic properties that are difficult generalize across regions [7–9]. As McPhillips et al. noted, over half of the studies on extreme events lack explicit definitions, and substantial inconsistencies exist across disciplines in terms of terminology, thresholds, and whether impacts are included [10]. They call for clearer and more consistent definitions to support cross-disciplinary understanding. Building on this, Alvre et al. (2024) reviewed over 2500 papers (2019–2023) and reached similar conclusions [11].

Vegetation phenology captures plant growth dynamics influenced by weather conditions [12,13]. It is highly sensitive to long-term climatic variations and is thus considered one of the best indicators of ecosystem dynamics [14]. Vegetation phenology can be assessed through the SOS (start of season), EOS (end of season), and GL (growing season length). Changes in vegetation phenology can significantly impact growth and reproduction patterns of species, which in turn influence material cycling and biodiversity within ecosystems and their essential ecosystem services, such as water regulation and soil conservation [15]. Vegetation phenology and growth are tightly linked through photosynthesis, providing cellular structure buildings. During photosynthesis, light is re-emitted from the chlorophyll molecules in leaves as a "side-effect", which is called Sun-induced chlorophyll fluorescence (SIF) [16,17]. The level of SIF indicates the plant's health, energy production power, and overall conditions within an ecosystem through climatic changes when monitored over time [18,19]. Balde et al. (2024) and Martini et al. (2022) investigated the effect of extreme heat events in Europe on evergreen broadleaved trees characterized with a relatively invariant canopy structure using field-scale SIF, and both found an inverted relationship between photosynthesis and fluorescence, showing a highly nonlinear protective mechanism of the plants from the adverse effects of high light intensity; they also found that both SIF and growth variations and their relationships depend on the temporal scale [20,21]. Against traditional measurements at small spatial and temporal scales, remote sensing-observed SIF with satellites provides valuable data for large-scale analyses.

The Mongolian Plateau (MP) is one of the world's largest plateaus and exhibits a predominantly semi-arid to arid climate regime [22,23]. Situated at the periphery of the East Asian monsoon weather system, this region demonstrates pronounced seasonality with prolonged cold winters and brief hot summers, coupled with marked precipitation variability [24,25]. These climatic features make the MP ecosystems highly sensitive to climate change, gradually becoming one of the most climate-sensitive regions in the world [26,27]. The MP has witnessed a marked escalation in both the occurrence and the severity of EWEs over recent decades, triggering substantial impacts on its fragile ecosystems [28–31]. EWEs such as extreme heat or cold wave, heavy precipitation, and various droughts have posed serious challenges to the stability of the ecosystems, accelerating soil erosion, vegetation cover loss, and soil moisture depletion, and potentially disrupting the normal phenological cycles of vegetation [32–34]. Previous research on MP vegetation phenology predominantly explored EWEs impacts on SOS and EOS, while their effects on GL remain understudied. Increasing evidence suggests that GL is directly related to the ecosystem carbon sequestration capacity, with a longer GL promoting plant growth and photosynthesis, thereby enhancing carbon fixation and storage [33,35–38]. Additionally, variations in GL are closely linked to water use efficiency, with a longer growing season facilitating more efficient soil water utilization and improving plant drought resistance [39–41]. In the context of climate change, alterations in GL duration, whether prolonged or shortened, could lead to changes

in ecosystem carbon sink function, thereby impacting the global carbon cycling and the climate system [42,43]. Therefore, investigating the response of GL, alongside SOS and EOS, to EWEs is crucial for understanding the relationship between plant growth cycles and climate change, and for providing scientific guidance in addressing climate change.

Event coincidence analysis (ECA) is a key methodology for evaluating vegetation responses to extreme climate impacts [7]. Unlike traditional statistical methods such as linear regression, which focus on examining the overall relationship between two variables, ECA specifically investigates the probability of concurrent occurrences of multiple events. This makes ECA particularly suited for exploring the direct response of GL to EWEs.

This study proposes a novel method for quantifying EWEs of heat, drought, cold, and wetness, and employs SIF data to evaluate ecosystem GL responses on short-term weather dynamics and long-term climate change. The study area is the MP with grasslands as the dominant ecosystem; however, the methodological framework is generalizable. The key objectives included (1) characterizing spatiotemporal patterns of grassland phenology; (2) quantifying coincidence probabilities between GL anomalies and EWEs using ECA; and (3) determining GL sensitivity gradients to EWE intensity variations.

## 2. Materials and Methods

### 2.1. Study Area

The MP is located in the central hinterland of Asia with a significant geographical extent on the Asian continent (Figure 1). Encompassing Mongolia and China's Inner Mongolia Autonomous Region, the MP extends between latitudes 37°46′ and 53°08′N, and longitudes 87°40′ and 122°15′E, with a total land area of approximately 3.5 million square kilometers.

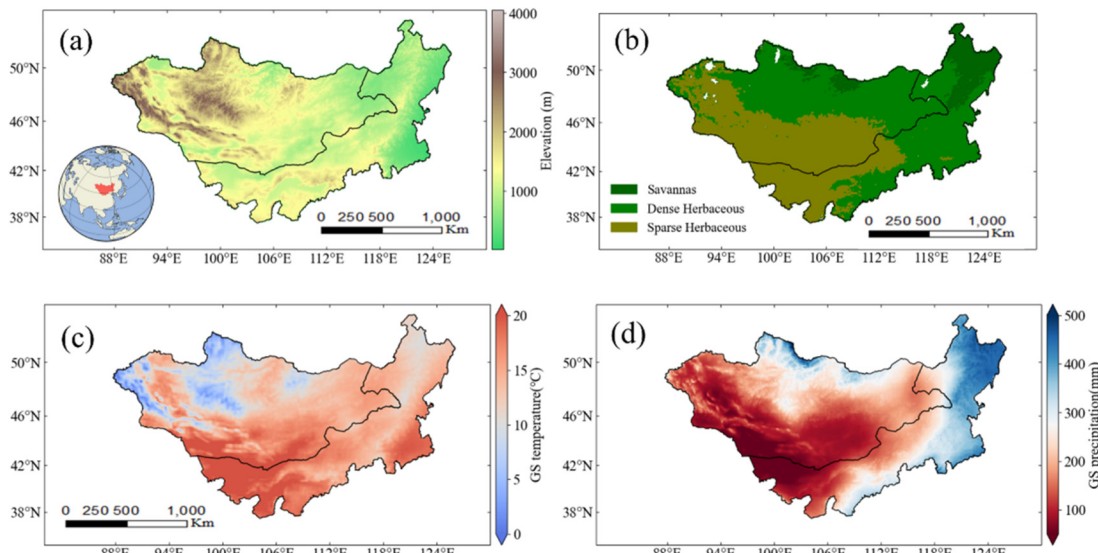

**Figure 1.** Mean elevation (**a**), land cover (**b**), temperature (**c**) and precipitation (**d**) of the Mongolian Plateau. The Mongolia–Inner Mongolia border is depicted by the line. Land cover classification derived from MODIS product (MCD12Q1 V006). Temperature and precipitation are based on 2001–2020 and refer to the growing season (GS) from April to September.

The MP is a typical plateau region, characterized by prominent mountain ranges in the west and north, where elevations are relatively high, whereas the eastern and southern parts are comparatively flat (Figure 1a). The dominant vegetation type is grassland, with overall vegetation coverage being relatively low due to the region's arid climate. The northeastern plateau supports savanna ecosystems, contrasting with sparse grasslands dominating

southwestern areas (Figure 1b). A marked north–south thermal gradient governs the region, with elevated southern temperatures and cooler northern zones (Figure 1c), shaped by topographic complexity and seasonal dynamics. The continental climate combines aridity, thermal limitations, and pronounced seasonality. Precipitation patterns show an east–south distribution (Figure 1d), with western–northern zones receiving minimal rainfall; over 60% of annual precipitation falls during summer months.

### 2.2. Datasets

### 2.2.1. Climate Datasets

This study utilized the ERA5 reanalysis temperature and precipitation dataset (European Centre for Medium-Range Weather Forecasts) to assess vegetation responses to EWEs. Selected for its robust quality control and temporal consistency, ERA5 supports the long-term analysis of ecosystem–climate interactions. Extreme temperature thresholds were derived from daily maximum/minimum values rather than averages, as temperature extremes exert stronger controls on vegetation physiology instead of mean conditions [4]. Cumulative precipitation during the growing season (April–September) was analyzed to capture hydrological impacts on vegetation dynamics. All weather variables were temporally aligned with the vegetation growing period.

### 2.2.2. SIF Vegetation Index Data

The SIF(CSIF) dataset used in this study was derived using a neural network model developed previously [44] and covered the period 2000–2022. This dataset has a temporal resolution of 4 days and a spatial resolution of 0.05° (ca. 4 km over the MP). To ensure spatial consistency with the meteorological data, the SIF data were resampled to a 0.1° resolution. The data were obtained from the National Tibetan Plateau Data Center of China (http://data.tpdc.ac.cn (accessed on 1 September 2023)).

### 2.2.3. Land Cover Classification

This study employed MODIS MCD12Q1 V006 (500 m resolution) land cover data, adhering to the International Geosphere-Biosphere Programme (IGBP) classification system. Vegetation was classified into three ecosystem types: forests (30–60% tree cover), dense herbaceous grassland (>60% herbaceous cover <2 m), and sparse herbaceous grassland (10–60% herbaceous cover <2 m). Non-vegetated areas (e.g., urban, water) were excluded. To ensure spatial consistency with the meteorological data, land cover layers were resampled to a 0.1° resolution via mode filtering. A 15-year land cover stability threshold (unchanged classification in ≥15 of the 20 years) was applied to minimize temporal variability impacts, enhancing reliability in assessing the vegetation–climate interactions.

### 2.3. Methods

### 2.3.1. Extraction of Grassland Phenology

This study utilized the PhenoFit R package [45] to extract phenological metrics from the SIF time series data. PhenoFit employs a weighted curve-fitting approach to reconstruct time series and offers multiple phenological extraction methods. The derivative approach (DER) was selected to identify the SOS (start of season) and EOS (end of season). Initially, quality control was performed on the input SIF time series to remove outliers (spike values) and missing values. To further reduce noise, the weighted Whittaker smoothing method [46] was applied as a preliminary curve-fitting step. PhenoFit effectively handles the irregular sampling characteristics of SIF data, making it suitable for datasets with varying temporal resolutions.

Subsequently, the first-order DER was used to compute the rate of change in the vegetation index, with the SOS and EOS identified as the time points corresponding to the

maximum positive and negative rates of change, respectively. The GL (growing season length) was then derived as the difference between the EOS and SOS. The DER method effectively captures vegetation growth dynamics and has been widely applied in phenological studies across diverse ecosystems [47]. Compared to traditional threshold-based methods, the derivative approach provides a more accurate representation of SIF change inflection points, thereby avoiding biases associated with arbitrary threshold selection.

### 2.3.2. Identification of Extreme Weather Events

This study examined four categories of extreme weather events: extreme heat, extreme cold, extreme wetness, and extreme drought. Primary metrics were established using monthly peak temperatures, monthly temperature minima, and cumulative monthly precipitation as diagnostic indicators. To address the relatively short duration of the ERA5 time series (2000–2022), which may limit the statistical robustness of extreme value estimation, we applied a $3 \times 3$ moving spatial window to increase the effective sample size for each grid cell. This method utilizes the spatial autocorrelation of climatic variables to improve the stability of threshold determination, especially in semi-arid and arid regions where spatial homogeneity is relatively high. Similar spatial aggregation approaches have been successfully applied in previous studies dealing with short time series of extreme weather events. To eliminate long-term trends from the time series of monthly climate variables during the growing season (April–September), we applied an ordinary least squares (OLS) regression to each pixel independently. This involved fitting a linear model of the climate variable (e.g., monthly maximum/minimum temperature or cumulative precipitation) as a function of the year. The residuals, obtained by subtracting the fitted values from the original data, were then used for standardizing anomalies. This detrending step allowed us to isolate interannual fluctuations from long-term changes, enabling a more robust detection of extreme climate events. Annual weather signatures were subsequently derived by aggregating monthly indicator values across the vegetative period. The detection of extremity thresholds employed a z-score transformation methodology. This standardized anomaly approach calculates deviations from climatic norms through the following formula: (observed value—multiannual mean)/standard deviation, enabling the quantitative evaluation of event severity. The identification framework commenced with detrending of monthly climatic variables, progressed through the temporal aggregation of growing season parameters, and ultimately implemented statistical standardization for extremity classification. The formula is as follows:

$$\lambda = \frac{X_i - mean(X)}{std(X)} \tag{1}$$

$\lambda$ represents the standardized anomaly, $X_i$ corresponds to the annual parameter value, $mean(X)$ denotes the multiannual mean, and $std(X)$ quantifies the interannual standard deviation. Extreme weather events are classified when temperature or precipitation anomalies exceed $\pm 1\, std(X)$ thresholds: values $\lambda \geq +1\, std(X)$ indicate extreme heat or pluviosity (positive deviations), whereas $\lambda \leq -1\, std(X)$ signifies extreme cold or aridity (negative deviations). The vegetation phenology analysis also adopts analogous thresholds, where standardized vegetation index anomalies below $-1\, std(X)$ are categorized as phenological shortening events, while those exceeding $+1\, std(X)$ denote lengthening events. This framework facilitates the quantitative evaluation of geographically distinct vegetation responses to divergent extreme climate regimes, revealing region-specific mechanistic linkages between climatic stressors and altered phenological dynamics. The systematic detection of phenological shifts under extreme conditions supports spatially explicit impact assessments of vegetation–climate interactions.

### 2.3.3. Event Coincidence Analysis

This study applied ECA to statistically evaluate the synchronization probability between extreme climatic events and vegetation phenological anomalies. The coincidence rate (CR, unitless) was calculated as the ratio of temporally concurrent event pairs (EWEs and GL anomalies) to the total EWEs recorded during the 2001–2021 observation period.

$$\text{CR} = \frac{Fre\left(\lambda_{veg} \forall t \text{when} \lambda_{climate}\right)}{Fre(\forall t \text{when} \lambda_{climate})} \tag{2}$$

where $\lambda_{veg}$ corresponds to vegetation phenological anomalies and $\lambda_{climate}$ to extreme weather events. The CR, ranging 0–1, quantifies vegetation–climate linkages, with elevated CR values reflecting stronger vegetation responses to weather extremes.

We investigated synchronous occurrences between the four EWEs (heat, cold, wetness, and drought) and vegetation GL anomalies. To distinguish statistically meaningful associations from random alignments, significance testing was conducted through temporal randomization. This significance test followed a permutation-based framework, as widely used in ECA applications [7], involving the preservation of the original GL anomaly chronology while shuffling EWEs timings, followed by recalculating CR values for 100 randomized permutations to establish baseline coincidence probabilities. Statistical significance was determined by comparing the original CR distribution against randomized counterparts via a $t$-test ($p < 0.05$). This rigorous validation confirmed that observed vegetation–climate event linkages exceeded chance-level fluctuations, ensuring the robustness of the detected relationships.

### 2.3.4. Sensitivity Analysis

In this study, absolute values of $\lambda_{veg}$ were applied to eliminate directional discrepancies among EWE types (e.g., opposing signs between extreme heat and cold anomalies). This normalization ensured the consistent measurement of their absolute impacts on vegetation phenology.

For each pixel across the study period (2001–2021), $\lambda_{veg}$ (in standard deviations) was scaled against the $\lambda_{climate}$ anomaly (in standard deviations) for each event type. The ratio was defined as follows:

$$\gamma = \frac{\lambda_{veg}}{|\lambda_{climate}|} \tag{3}$$

The ratio $\gamma$ between these variables functions as a vegetation sensitivity indicator to climatic extremes. A positive $\gamma$ value signals enhanced vegetation growth during an extreme climate event relative to baseline conditions, whereas a negative value reflects growth suppression. This metric offers a direct quantitative measure of vegetation–climate interaction intensity, clarifying the proportional response of ecosystems to climatic extremes.

## 3. Results

### 3.1. Spatial Distribution and Interannual Variation of Vegetation Phenology in the Mongolian Plateau

The spatial patterns of vegetation phenology (SOS, EOS, GL) across the MP during 2001–2020 revealed pronounced regional heterogeneity (Figure 2). The SOS exhibited marked latitudinal contrasts: mid-April onset in southwestern areas versus late April–early May in the northeast, with central and southwestern regions delayed to late May–June (Figure 2a). The EOS displayed a southwest–northwest decreasing gradient, with a mean value of day 260 in southern Inner Mongolia versus day 220 in western arid areas (Figure 2b). Its distribution followed hydrothermal gradients, linearly increasing from day 230 ($-1$ °C) to day 250 (19 °C). For GL, the northeastern humid regions exhibited a longer growing

season, with an average duration of 110 days. In contrast, the drier southwestern regions had a significantly shorter growing season, averaging only 50–70 days. In other areas, GL generally ranged between 80 and 100 days (Figure 2c).

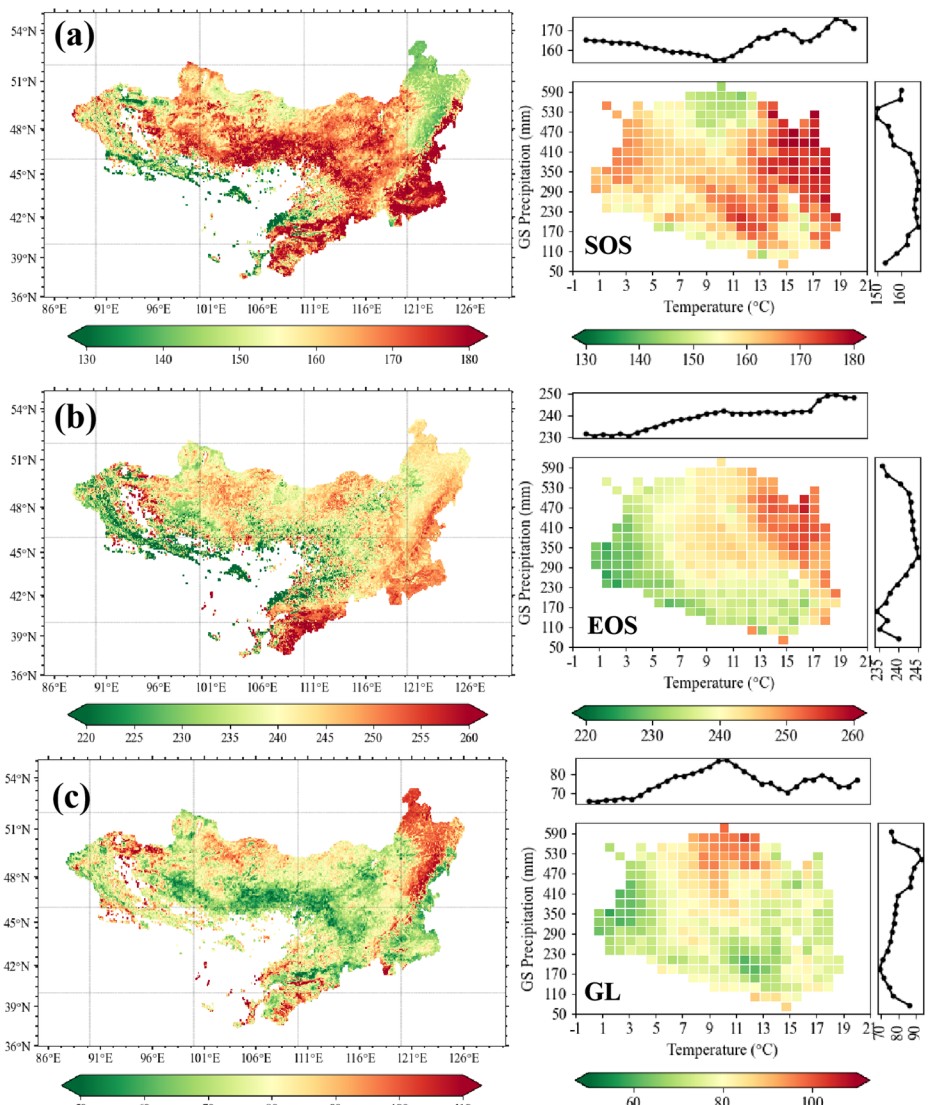

**Figure 2.** Spatial distribution of vegetation phenology (unit of days) on the Mongolian Plateau and its variation along climatic gradients. Start (SOS; **a**) and end of the growing season (EOS, **b**) and growing season length (GL, **c**) are presented in two panels: left is multi-year (2001–2020) mean, right is annual variations along regional growing season (GS) temperature and precipitation gradients.

A further analysis of the interannual trends in vegetation phenology revealed significant regional variations across the MP (Figure 3). During the period 2001–2020, the SOS exhibited a distinct spatial pattern. In the central and northern regions of the plateau, the SOS generally showed an increasing trend, indicating a delayed onset of vegetation growth. In contrast, the eastern and southeastern regions exhibited a decreasing trend, suggesting an earlier start of the growing season (Figure 3a). For EOS, changes were relatively minor across most of the study area (Figure 3b). Apart from certain central regions, where the EOS increased at a rate exceeding 0.5 days per year, 70.2% of the study area experienced only marginal changes, with EOS trends falling within the range of −0.5 to 0.5 days per year. The GL trend map revealed that 26.2% of the study area experienced a significant increase in growing season length (>0.5 days per year), predominantly concentrated in the southeastern region, indicating a notable extension of the phenological growing season

(Figure 3c). Conversely, 17.8% of the study area showed a decreasing GL trend exceeding −0.5 days per year, primarily occurring in the arid regions of the MP, suggesting a shortening of the growing season in these areas.

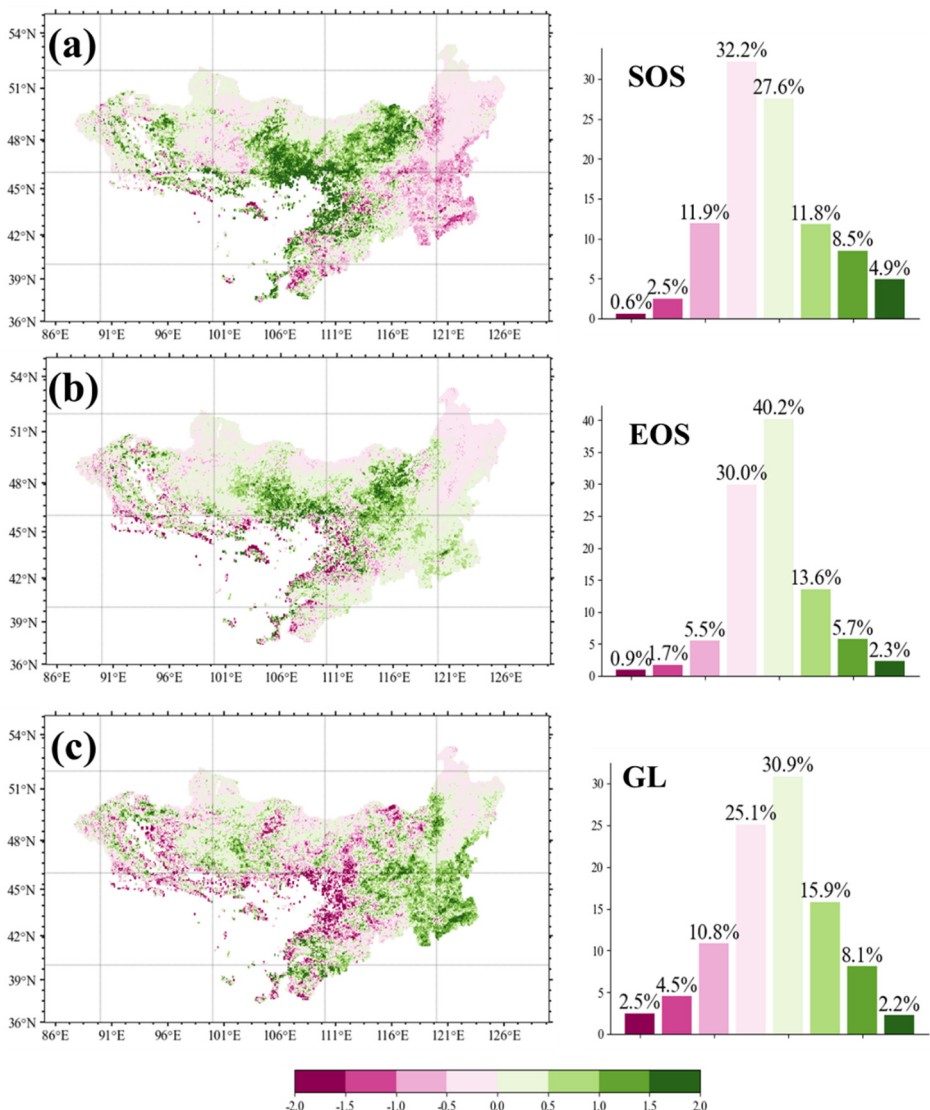

**Figure 3.** Interannual variation trend in phenology (unit of days). Left figure shows the distribution statistics within each variation trend interval for start (SOS; **a**) and end of the growing season (EOS; **b**) and the length of the growing season (GL; **c**). Bars on the right show corresponding %-wise contribution.

### 3.2. Coincidence Analysis of GL and EWEs

We quantified CR between GL anomalies and four EWEs (extreme heat, cold, wet, drought), assessing divergent responses of positive versus negative GL anomalies to climatic extremes (Figure 4). Across the MP, negative GL anomalies exhibited a strong association with all EWEs, with 94% of pixels displaying CR > 0.2, and 8.2% of pixels showing an extremely high coincidence rate (CR > 0.8). These high-coincidence regions were predominantly in the arid and low-precipitation areas in the southeastern MP (Figure 4a). For positive GL anomalies, 95.5% the study area exhibited a CR > 0.2, and 9% of vegetated areas had a CR exceeding 0.8 with EWEs. However, the spatial distribution of high CR values differed from that of negative anomalies, with high-coincidence regions primarily occurring in relatively humid areas in the western and eastern MP (Figure 4b). Overall, in response to extreme climate events, negative GL anomalies exhibited higher coincidence

rates than positive GL anomalies, indicating that reductions in growing season length were more strongly associated with weather extremes than growing season extensions.

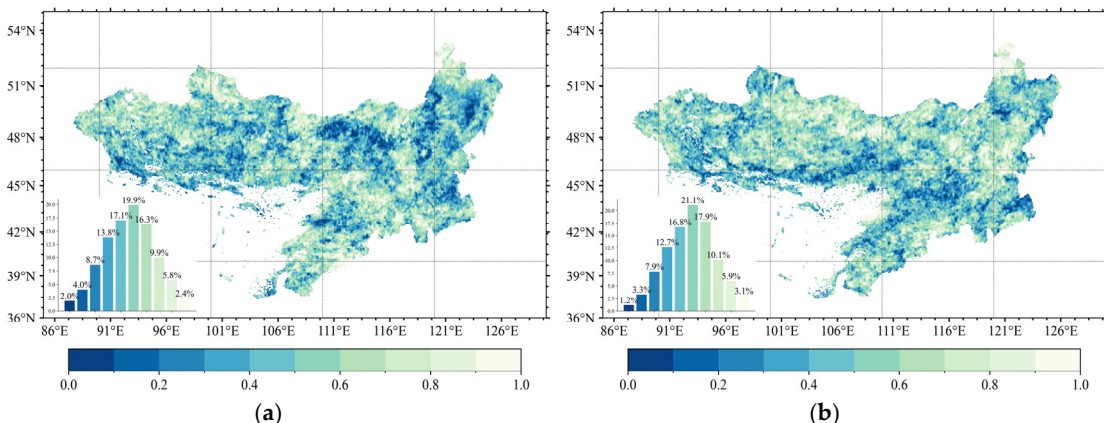

**Figure 4.** Spatial patterns of coincidence rates between growing season length anomalies (negative, **a**, and positive, **b**) and extreme weather events across the Mongolian Plateau. Bars in sub-plots show corresponding %-wise contribution of each CR class.

We further examined the CR values between GL anomalies and individual extreme weather event types, providing a more detailed spatial assessment of their relationships (Figure 5). Across the MP, the dominant extreme weather event type influencing negative GL anomalies varied by region (Figure 5a). In the humid, high-vegetation coverage areas of the northeast, negative GL anomalies were primarily associated with extreme wet conditions. In contrast, in the southern and western regions, where long-term low precipitation is prevalent, extreme drought was the dominant driver of negative GL anomalies. In parts of the northwestern and northern MP, extreme low temperatures played a leading role, whereas in the central MP, extreme high temperatures emerged as the primary climatic factor influencing negative GL anomalies. For positive GL anomalies, the dominant extreme weather event types also exhibited regional variations (Figure 5b). Moreover, spatial patterns of extreme weather impacts diverged between positive and negative GL anomalies, indicating that EWEs may modulate opposite to GL, depending on regional climatic conditions.

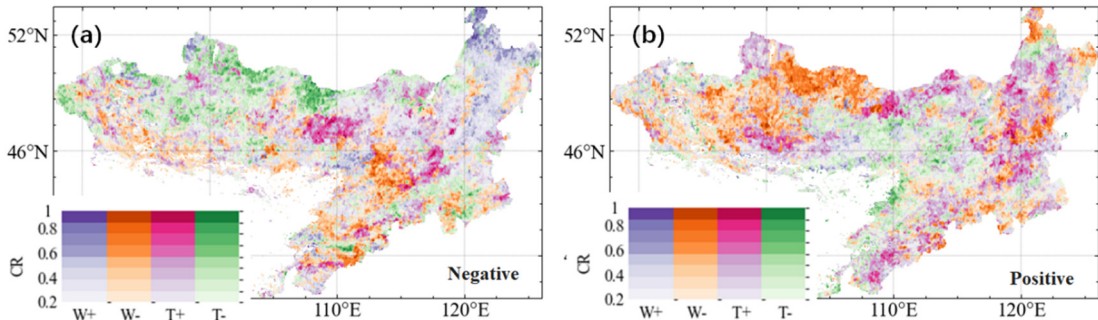

**Figure 5.** Spatial patterns of coincidence rates (CRs) between growing season length (GL) anomalies (negative, **a**, and positive, **b**) and extreme climate events. Symbols W−, W+, T+, and T− denote extreme drought, extreme humidity, extreme high temperature, and extreme low temperature, respectively. Colour legend of CR values is shown in the lower left corner of each plot.

Our analysis revealed that EWEs predominantly drive GL anomalies across the MP (Figure 6), accounting for 83.7% of negative and 87.4% of positive anomalies. For negative GL anomalies, contributions from extreme drought, wetness, heat, and cold were 18.5, 27, 20.5, and 17.7%, respectively. Their corresponding CR values ranked as follows: extreme

wetness (0.39) > heat (0.38) > drought (0.37) > cold (0.36). Positive GL anomalies exhibited distinct EWE contributions: drought (15.1%), wetness (27.7%), heat (25.4%), and cold (19.2%), with CR values decreasing from wetness (0.42) to heat (0.38), drought (0.36), and cold (0.35).

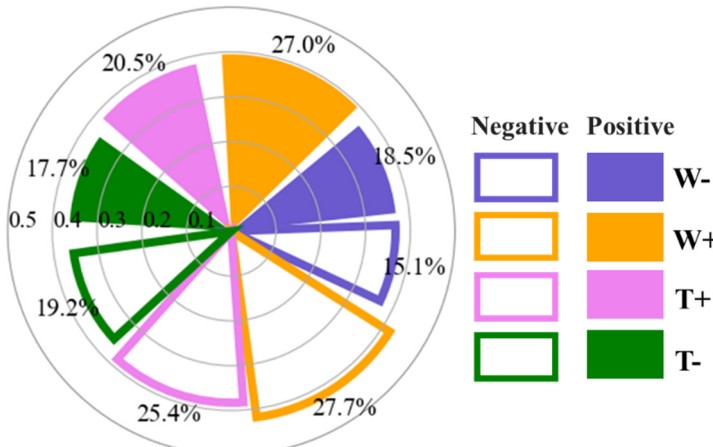

**Figure 6.** Contribution metrics of individual EWEs to GL anomalies (negative/positive), quantified by dominant area percentage and coincidence rate (CR), analyzed for statistically valid pixels ($p < 0.05$). Symbols W−, W+, T+, and T− denote extreme drought, extreme wetness, extreme high temperature, and extreme low temperature, respectively.

### 3.3. Dependence of the Coincidence Rate on Regional Background Hydrothermal Conditions

In this section, we examine spatial patterns of CRs for four EWEs (Figure 7), revealing distinct thermal dependencies. Extreme heat displayed a rising CR with higher growing season temperatures, peaking (CR = 0.31) at 21 °C, while extreme cold events exhibited declining CR trends despite fluctuations, inversely correlated with thermal gradients. A similar pattern was found for extreme drought and extreme wet conditions. As precipitation levels increased, areas with higher annual average rainfall during the growing season exhibited a progressive decline in the CR of extreme drought. A distinct stepwise downward gradient was observed, with CR values remaining low in regions where annual growing season precipitation exceeded 200 mm. For extreme wet conditions, a reverse trend was evident. As precipitation levels increased, regions became increasingly influenced by extreme wet events, leading to a gradual rise in CR values. A clear gradient shift was observed, with regions experiencing more than 250 mm of annual precipitation during the growing season, showing higher susceptibility to extreme wet conditions. In these areas, CR values consistently exceeded 0.2, indicating a greater influence of extreme wet events on vegetation phenology.

Similarly, we analyzed the dependence patterns of the coincidence rate (CR) for the four types of extreme weather events on background hydrothermal conditions in relation to positive GL anomalies (Figure 8). Notably, whether under a precipitation gradient or a temperature gradient, the CR of extreme weather events exhibited distinct trend variations in response to background hydrothermal conditions. However, these trends followed an opposite pattern compared to those observed in negative GL anomalies. Specifically, under the temperature gradient, extreme high-temperature events exhibited higher CR values in regions with lower annual average temperatures during the growing season, reaching a maximum of 0.3. In contrast, for extreme low-temperature events, CR values showed strong fluctuations, similar to the pattern observed in negative GL anomalies. Along with precipitation gradients, extreme drought CRs increased with higher annual precipitation but remained lower in arid zones. Extreme wet conditions exhibited a reverse trend, with CRs peaking in drier regions and declining in humid zones.

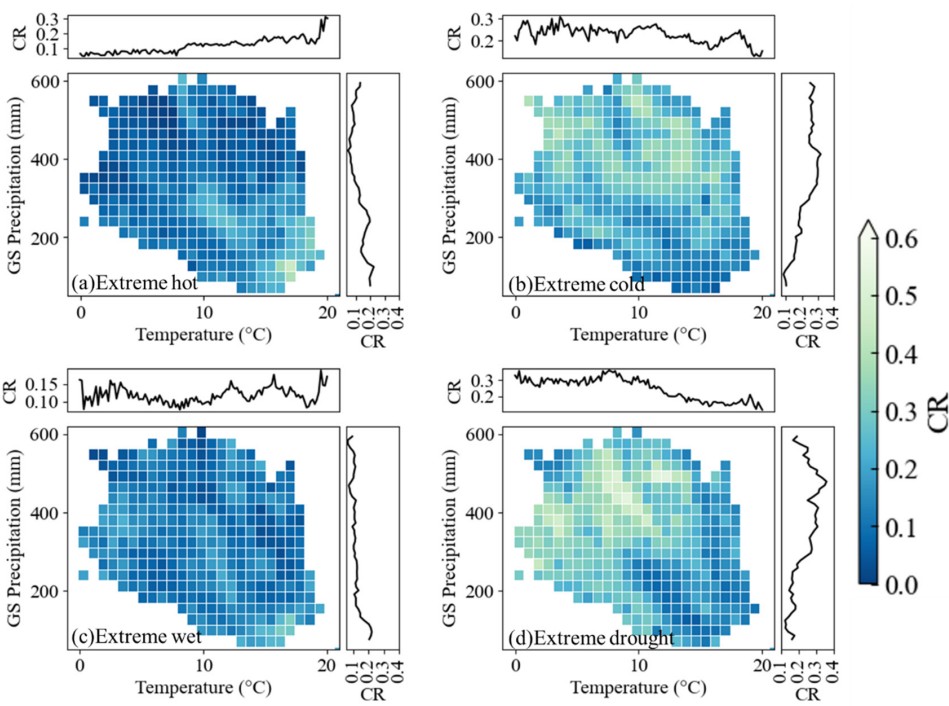

**Figure 7.** Coincidence rate (CR) distribution between negative growing season length anomalies and extreme weather events across regional hydrothermal gradients (regional growing season (GS) temperature and precipitation). Top and right sub-plots around each plot depict mean CR per 0.5 °C thermal or 50 mm precipitation interval, respectively.

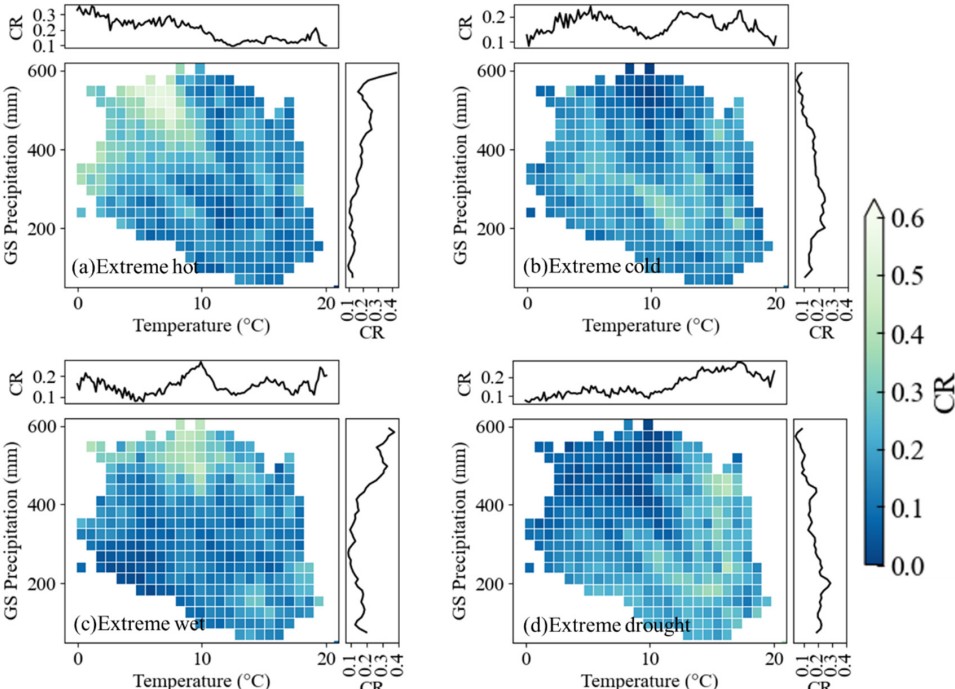

**Figure 8.** Coincidence rate (CR) distribution between positive gradients (regional growing season (GS) temperature and precipitation). Top and right sub-plots around each plot depict mean CR per 0.5 °C thermal or 50 mm precipitation interval, respectively.

In summary, the analysis reveals the distribution characteristics of CR values of GL anomalies caused by extreme weather based on local background hydrothermal conditions in the MP. The study found that when extreme high temperature or extreme wetness events occur in areas with a higher annual mean temperature or higher precipitation during the

growing season, the possibility of GL shortening is higher. Similarly, when extreme drought events occur in areas with less precipitation, the possibility of GL shortening is higher. On the contrary, in areas with a lower annual mean temperature or less precipitation during the growing season, the possibility of GL lengthening is higher when extreme high temperature or extreme wetness occurs.

### 3.4. Sensitivity of GL to Extreme Climate Events During the Growing Season

Vegetation GL sensitivity to EWEs displayed marked spatial heterogeneity across the MP (Figure 9). Extreme cold events induced GL suppression (negative sensitivity) in 56.4% of pixels, with the strongest reductions clustered in northwestern regions. Conversely, extreme heat triggered a widespread GL enhancement (positive sensitivity), spatially mirroring cold-event impacts. Extreme wetness promoted GL in 42.5% of pixels, particularly in the northeast, whereas drought caused GL shortening in 43.2% of areas, highlighting water limitations on phenological duration. These results demonstrate region-specific vegetation responses to EWEs, with opposing event types (e.g., heat vs. cold, wet vs. drought) generating contrasting GL anomalies within shared geographic contexts.

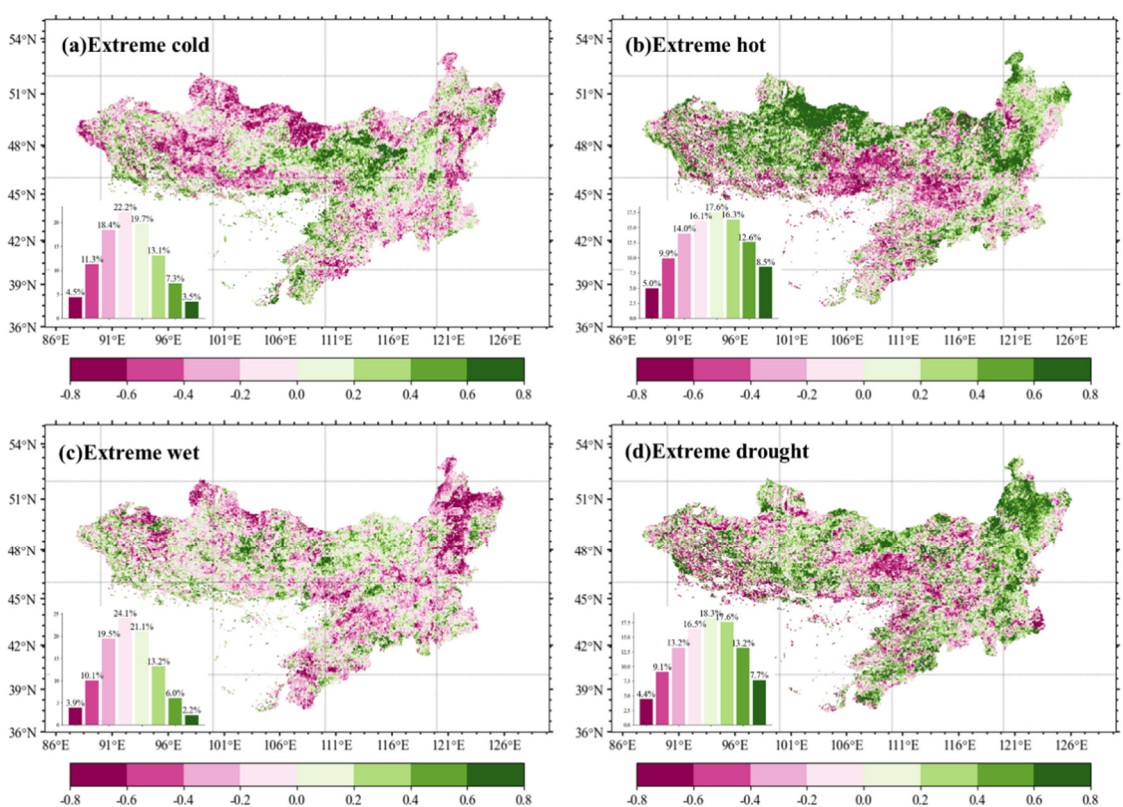

**Figure 9.** Vegetation growing season length (GL) anomaly sensitivity to four different extreme weather events (EWEs). Bars in sub-plots show corresponding %-wise contribution of each event type.

The subsequent analysis demonstrates that the vegetation type-specific sensitivity to EWEs aligns with regional hydrothermal gradient-driven CR variations (Figure 10). For extreme low-temperature events, when growing season temperatures are below 12 °C, extreme cold conditions have an abnormal suppressive effect on the vegetation growing season length (GL), leading to GL shortening. However, as temperature increases, the influence of extreme low-temperature events shifts from an inhibitory to a promotive effect on GL extension. In contrast, for extreme high-temperature events, the sensitivity of vegetation phenology exhibits a symmetrical distribution relative to that of extreme low-temperature events. When growing season temperatures exceed 12 °C, extreme heat

conditions promote GL extension, whereas at temperatures below 12 °C, GL is shortened by extreme high temperatures. For extreme wet and extreme dry events, sensitivity patterns follow the precipitation gradient. Extreme wet conditions exhibit an increasingly inhibitory effect on GL as precipitation increases, whereas extreme drought conditions have the opposite effect, progressively enhancing GL extension as precipitation levels decline.

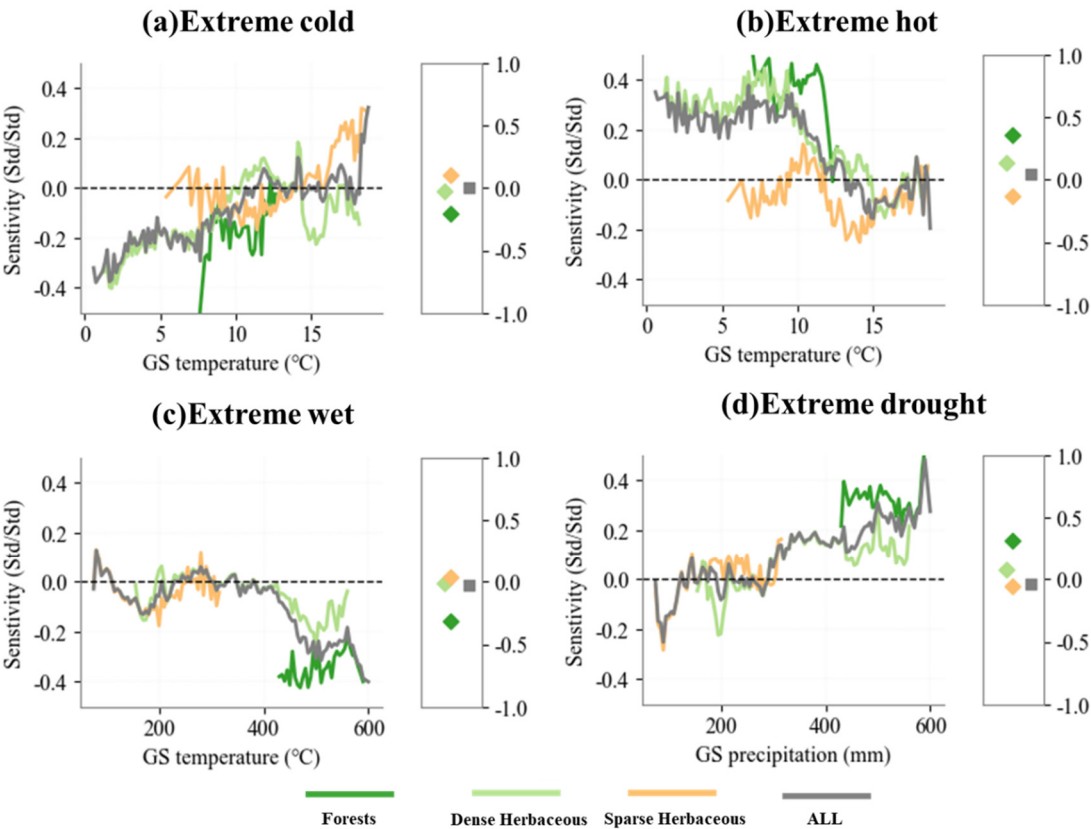

**Figure 10.** Distribution of growing season length (GL) sensitivity to four different extreme weather events for different vegetation types along the temperature and precipitation gradients in the Mongolian Plateau. Box on the right of each plot shows the mean sensitivity for each vegetation type.

## 4. Discussion

### 4.1. Hydrothermal Modulation of Vegetation Thermal-Hydraulic Sensitivity

This study demonstrates that regional hydrothermal climatic conditions modulate EWE-driven GL variations of grassland ecosystems across large geographical areas such as the MP. Grassland GL sensitivity to extreme wet events was primarily controlled by local hydrological regimes, whereas thermal extremes exerted temperature-dependent impacts governed by baseline climatic conditions.

First and more obviously, the sensitivity of vegetation to extreme wet events is closely related to local hydrological conditions. In humid regions, excessive water can inhibit plant growth, leading to root oxygen deficiency and limited photosynthetic activity [48,49], which may also lead to the accumulation of toxic substances such as alcohols and aldehydes, further disrupting the normal physiological functions of plants [50]. In contrast, vegetation in dry regions is susceptible to damage caused by drought [51]. This is because vegetation in arid regions is more dependent on water availability during growth, and even short-term water deficits can cause physiological damage to plants.

Secondly, in warmer regions, GL was more susceptible to damage from extreme heat, whereas in colder regions, it was vulnerable to extreme cold. The SIF data showed higher values in the warm, humid regions where heat extends the growing season, and lower

values in the arid/cold regions where heat induces drought stress or exceeds thermal tolerance (Figures 1 and 2). Plants in warmer regions are adapted to high temperatures, but temperatures exceeding their heat tolerance thresholds reduces photosynthesis efficiency significantly [52]. High temperatures cause damage to photosynthetic enzymes, particularly the Rubisco enzyme, reducing carbon fixation and inhibiting photosynthesis [53]. Extreme heat promotes SIF because plants experience saturation of nonphotochemical quenching, causing a change in the allocation of energy dissipation pathways towards SIF (15). Low temperatures cause plant cell walls and membranes to lose elasticity, and the water inside the cells freezes, leading to structural damage and disrupting normal metabolic functions [54]. Low temperatures also reduce the efficiency of photosynthesis, affecting chloroplast function, particularly the efficiency of photochemical electron transport and ATP synthesis, which limits plant growth [55].

### 4.2. Divergent Ecosystem Adaptations to Extreme Drought Under Uniform Precipitation

According to the land cover classes, forest ecosystems had greater drought resistance than grasslands under equivalent precipitation conditions. This difference is closely related to the morphological and physiological adaptations of vegetation [8,56]. Woody plants have deep root systems, allowing for the effective utilization of deep soil moisture, while thickened xylem conduits and leaf cuticles help reduce transpiration losses [57,58]. Particularly in mixed forests, as species diversity increases, drought resistance also improves, which is closely linked to the complementary water use strategies of different tree species [59]. In contrast, root systems of herbaceous plants in grasslands are relatively shallow compared to forests, making them more dependent on surface soil moisture, resulting in weaker drought resistance [57,60].

The influence of temperature gradients on plant growth shows significant differentiation. Compared to woody plants, herbaceous plants exhibit growth advantages in cold environments. Studies have shown that the optimal photosynthetic temperature threshold for herbaceous plants is lower than that of woody plants [61]. This adaptation is attributed to their short life cycle characteristics, enabling them to maintain metabolic activity at low temperatures by adjusting the concentration of osmotic regulators, such as proline [62,63]. Conversely, woody plants adapt to high temperatures by providing canopy shading, which creates a microclimate and reduces under-canopy temperatures [64]. The increase in Rubisco enzyme activity induced by high temperatures enhances photosynthetic efficiency, allowing woody plants to better tolerate heat [65].

### 4.3. Methodological Contributions and Perspectives

This study has several limitations that warrant consideration. First, the effects of human activities such as grazing intensity and land-use practices are important factors that affect the vegetation phenology [66–69]. Overgrazing weakens the grassland's buffering capacity against extreme weather events by damaging vegetation cover and altering the soil microbial community structure [70,71]. Su et al. investigated the effects of grazing and vegetation type on community characteristics and ecosystem functions, indicating that grazing reduces the cover, height, species richness, and aboveground biomass in meadow and typical grasslands, but has little impact on desert grasslands [31]. Wang et al. (2023) analyzed a grazing dataset from 114 counties on the Tibetan Plateau in conjunction with SIF data to assess the impact of grazing intensity on vegetation phenology. Their study found that the sensitivity of the SOS and EOS to grazing intensity decreased as grazing intensity increased. Reducing grazing intensity enhanced soil moisture, resulting in an earlier SOS under moderate grazing levels in spring [72]. Therefore, different grazing intensities have

varying effects on plant phenology, which in turn influences the GL's responses to extreme climate events.

Secondly, this study primarily focused on the impacts of independent EWEs, neglecting the combined effects of multiple EWEs. The impact of compound events on ecosystems can be more severe than that of individual events. For instance, the co-occurrence of drought and heatwave events exert a more significant and pronounced destructive effect on natural ecosystems compared to isolated extreme heat or drought events [73]. Zhou et al. (2024) assessed the impact of Compound Drought and Heatwave Events (CDHEs) on vegetation across climate zones and ecosystems from 1993 to 2020. They found that vegetation was most affected by CDHEs lasting 5 to 9 days, with an expanded geographical extent and decreased lag time [74]. Future studies should prioritize examining the synergistic impacts of compound extreme climatic events on GL, particularly under intensifying global climate variability.

In addition to these limitations, our study also presents several conceptual and methodological innovations that address gaps in previous research. Most existing studies have focused on phenological responses to long-term climatic trends or individual extreme events, often treating them in isolation. In contrast, we employed event coincidence analysis (ECA) in combination with SIF data to systematically quantify the probability of growing season length (GL) anomalies, coinciding with distinct types of EWEs across hydrothermal gradients. Unlike traditional regression-based approaches, ECA captures short-term, discrete responses, providing a novel event-based perspective on vegetation sensitivity. Furthermore, while previous research has largely emphasized the effects of extreme drought and heat, our findings reveal the unexpectedly large and spatially heterogeneous influence of extreme wetness—both in shortening and extending the growing season, depending on regional hydrological conditions. This highlights the complex and often underestimated ecological role of moisture surplus in shaping ecosystem phenology. Finally, by applying a z-score-based framework to define EWE thresholds from detrended meteorological time series, our study offers a standardized and ecologically relevant method to detect climatic extremes and assess vegetation responses. This directly addresses a methodological gap highlighted by Walsh et al. (2020), who pointed out that existing observation and modeling approaches remain insufficient—particularly in high-latitude and arid regions, where threshold definitions and ecological impact assessments are still scarce [9].

## 5. Conclusions

This study systematically investigated the response of growing season length (GL) to extreme weather events (EWEs) on the Mongolian Plateau (MP) from 2000 to 2022, using solar-induced fluorescence (SIF) data and event coincidence analysis (ECA). It is among the very few studies to define EWEs for geo-environmental investigations statistically based on long time series of ERA5 meteorological data through initial detrending extremity threshold detection through z-score transformation. The coincidence analysis revealed that 83.7% of negative GL anomalies and 87.4% of positive anomalies coincided with at least one type of EWE, emphasizing their dominant role. Extreme wetness contributed the most to GL variations (27.0% of negative and 27.7% of positive anomalies), followed by extreme heat (20.5% and 25.4%, respectively), highlighting the differential roles of moisture- and temperature-related stressors across hydrothermal gradients in the MP.

The results further highlighted that negative GL anomalies were more strongly associated with EWEs than positive anomalies, with extreme heat and extreme wet events significantly contributing to GL shortening in warmer and wetter regions, respectively. Conversely, extreme cold and extreme drought events played a dominant role in drier and colder regions, leading to varied GL responses. Additionally, we observed that hydrother-

mal conditions modulate vegetation sensitivity, with warmer regions more susceptible to heat stress and drier regions more vulnerable to drought impacts.

In addition to quantifying the dominant influence of EWEs on phenological variability, our study also contributes conceptually and methodologically by introducing a standardized, event-based framework that improves the detection of short-term vegetation responses to climatic extremes. These insights provide a valuable foundation for future ecological modeling and risk assessment in arid and semi-arid regions. Future research should further explore the compound effects of multiple extreme events and anthropogenic influences, such as grazing and land-use change, to better understand the resilience and vulnerability of ecosystems under intensified climate variability.

**Author Contributions:** Conceptualization, W.Z., K.M. and S.L.; methodology, WZ, K.M. and S.L.; software, W.Z. and Q.G.; validation, W.Z. and S.L.; formal Analysis, W.Z.; investigation, W.Z., Q.G., G.W., K.M. and S.L.; resources, W.Z.; data curation, W.Z., Q.G. and G.W.; writing—original draft preparation, W.Z.; writing—review and editing, W.Z., Q.G., G.W., K.M. and S.L.; visualization, W.Z.; supervision, K.M. and S.L.; project administration, S.L.; funding acquisition, S.L. All authors have read and agreed to the published version of the manuscript.

**Funding:** This research was funded by the National Key Research & Development Plan (2024YFF1306101) and the National Natural Science Foundation of China (32171555, 31961143022).

**Data Availability Statement:** Data will be made available on request.

**Acknowledgments:** The authors would like to thank the editors and two anonymous reviewers for their thoughtful comments, which improved the quality of this paper.

**Conflicts of Interest:** The authors declare no conflicts of interest.

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
