# Peer review of "Long-Term Impact of Extreme Weather Events on Grassland Growing Season Length on the Mongolian Plateau"

_remotesensing, doi:10.3390/rs17091560_

Round 1
Reviewer 1 Report
Comments and Suggestions for Authors
The abstract says that this paper provides new insights, but upon reading it, nothing new seems to be presented that is different from previous studies. Both the abstract and conclusions describe the impacts of extreme weather events on growing season length in qualitative terms. It is recommended that quantified contribution values be included in these sections. Furthermore, comparing the magnitude of the quantified contributions with findings from other studies would enhance the depth of the discussion section.
This manuscript aims to explore the grassland growing season length responses long-term of extreme weather events in the MP. The study has significant research value, and the manuscript is well-organized. However, there are some problems with the manuscript that need more attention.
Line 40-43: “These findings emphasize the importance of regional weather variability and climate characteristics in shaping vegetation phenology and provide new insights into how weather extremes impact ecosystem stability in semi-arid and arid regions.”
The abstract says that this paper provides new insights, but upon reading it, nothing new seems to be presented that is different from previous studies. Therefore, it is recommended that the new findings, which are different from previous studies, be further compared with previous studies in the discussion chapter, and the new insights proposed in this manuscript be more explicit.
Line 180-194: “2.3.2 Identification of extreme weather events”.
This manuscript used long time series (2000-2022) ERA5 meteorological data to identify the extreme weather events. At least 30 years of climate data is often needed to identify extreme weather events. Is it reasonable to rely on 20 years of data alone?
Both the abstract and conclusions describe the impacts of extreme weather events on growing season length in qualitative terms. It is recommended that quantified contribution values be included in these sections. Furthermore, comparing the magnitude of the quantified contributions with findings from other studies would enhance the depth of the discussion section.
Reviewer 2 Report
Comments and Suggestions for Authors
Review Comments for remotesensing-3564045
Long-Term Impact of Extreme Weather Events on Grassland Growing Season Length on the Mongolian Plateau
1. Overall Evaluation
This study evaluated grassland ecosystem GL responses on short-term weather dynamics and long-term climate change. The key objectives included: (1) characterizing spatiotemporal patterns of grassland phenology; (2) quantifying coincidence probabilities between GL anomalies and EWEs using ECA; and (3) determining GL sensitivity gradients to EWE intensity variations. The selected topic holds significant scientific relevance, with a relatively novel methodology, and the conclusions provide reference value for understanding the stability of semi-arid ecosystems. However, certain details require further refinement.
2. Possible Problems and Recommendations
(1) In Section 2.3.2:
"To mitigate the influence of longitudinal trends, an initial data processing phase removed long-term trends from growing season data (April-September) through a detrending procedure."
The specific method or steps of the "detrending procedure" are not clearly stated. Detrending typically involves removing linear or nonlinear trends from time-series data using techniques such as linear regression, polynomial fitting, or moving averages. Clarifying the chosen method and its implementation is critical for reproducibility.
(2) In Section 2.3.3:
The explicit description of the Event Coincidence Analysis (ECA) significance test is missing. The specific statistical method (e.g., permutation tests, p-value thresholds) and its theoretical basis (e.g., reference to established frameworks like those in climate science or ecological studies) should be provided. If existing methods are adopted, appropriate citations to foundational literature (e.g., references on ECA in ecological applications) are necessary.
(3) Figures:
Figure 2a: Text in the left panel is partially truncated, affecting readability. Ensure labels and annotations are fully visible.
Figure 3: The captions for the left and right panels are reversed. Verify alignment between figure descriptions and visual content to avoid misinterpretation.
